# A cytochrome P450 CYP87A4 imparts sterol side-chain cleavage in digoxin biosynthesis

Emily Carroll [1,2], Baradwaj Ravi Gopal [1,2], Indu Raghavan[1], Minakshi Mukherjee [1] & Zhen Q. Wang [1] ✉

Digoxin extracted from the foxglove plant is a widely prescribed natural product for treating heart failure. It is listed as an essential medicine by the World Health Organization. However, how the foxglove plant synthesizes digoxin is mostly unknown, especially the cytochrome P450 sterol side chain cleaving enzyme (P450$_{scc}$), which catalyzes the first and rate-limiting step. Here we identify the long-speculated foxglove P450$_{scc}$ through differential transcriptomic analysis. This enzyme converts cholesterol and campesterol to pregnenolone, suggesting that digoxin biosynthesis starts from both sterols, unlike previously reported. Phylogenetic analysis indicates that this enzyme arises from a duplicated cytochrome P450 *CYP87A* gene and is distinct from the well-characterized mammalian P450$_{scc}$. Protein structural analysis reveals two amino acids in the active site critical for the foxglove P450$_{scc}$'s sterol cleavage ability. Identifying the foxglove P450$_{scc}$ is a crucial step toward completely elucidating digoxin biosynthesis and expanding the therapeutic applications of digoxin analogs in future work.

Cardiac glycosides extracted from the foxglove plant *Digitalis lanata* have been used for treating congestive heart failure since 1785[1]. Digoxin, a widely prescribed cardiac glycoside, is listed as an essential medicine by the World Health Organization[2]. About 400,000 patients are prescribed digoxin in the United States, making it one of the most prescribed plant natural products[3]. Recent research has broadened the medicinal applications of cardiac glycosides for treating viral infection, inflammation, cancer, hypertension, and neurodegenerative diseases[4–11].

Due to the prominence of digoxin in medicine, the study of cardiac glycoside biosynthetic pathways dates back to the 1960s. Radiolabeling studies suggested cholesterol as the precursor for digoxin[12]. While this is generally accepted, controversies remain since cholesterol is a minor sterol in plants. The exact biosynthetic pathway of digoxin remains enigmatic half a century after the initial work[13]. The hypothetical cardiac glycoside biosynthetic pathway starts with cholesterol, which undergoes nine enzyme-catalyzed steps to digoxigenin, the aglycone of digoxin[13]. Currently, the only known enzymes in the pathway are 3$\beta$-hydroxysteroid dehydrogenase (3$\beta$HSD) and

progesterone-5$\beta$-reductase (P5$\beta$R and P5$\beta$R2)[14–16]. The first and rate-limiting enzyme, cytochrome P450 sterol side chain cleaving enzyme (P450$_{scc}$), along with all other enzymes, has not been identified yet[13,17]. The foxglove P450$_{scc}$ is thought to convert cholesterol to pregnenolone through a reaction identical to mammalian P450$_{scc}$, catalyzing the rate-limiting step in animal steroid hormone synthesis[18]. However, the plant P450$_{scc}$ has not been isolated and characterized since its first description by Pilgrim in 1972[19]. Hence, the direct sterol precursor for digoxin biosynthesis remains ambiguous. Indirect evidence suggests that phytosterols, including campesterol, stigmasterol, and sitosterol, may also be precursors for digoxin[12,19–23].

In this study, we utilized a high-quality transcriptome of *D. lanata* to identify the foxglove P450$_{scc}$. Characterizing the foxglove P450$_{scc}$ validated its sterol cleaving activity in tobacco and yeast. Investigating the foxglove P450$_{scc}$ substrate preference uncovered the identity of sterol precursors for digoxin biosynthesis. Phylogenetic analysis suggests that this enzyme evolved from the CYP87A family. Protein modeling and mutagenesis revealed critical amino acids for foxglove P450$_{scc}$'s sterol-cleaving activity. The foxglove P450$_{scc}$ is the first plant

[1]Department of Biological Sciences, University at Buffalo, the State University of New York, Buffalo, NY, USA. [2]These authors contributed equally: Emily Carroll, Baradwaj Ravi Gopal. ✉e-mail: zhenw@buffalo.edu

P450$_{scc}$ identified and does not share substantial homology with the animal P450$_{scc}$.

## Results

### Transcriptome assembly and annotation

Total RNA from leaf and root tissues, including three biological replicates and two technical replicates from each tissue, were pooled to generate a reference transcriptome of *D. lanata*. We performed de novo assembly of the transcriptome from 173,448,870 Illumina raw reads with an average length of ~100 bp. The assembled transcriptome contains 317,983 transcripts with an N50 of 1712 bp (Table 1, Supplementary Fig. 1). The Benchmarking Universal Single-Copy Orthologs (BUSCO) score for the transcriptome was 94.6%, indicating that the transcriptome was near complete. A total of 190,755 transcripts at least 300-bp long were annotated using publicly available databases, including the NCBI non-redundant protein database (nr) and the Uni-Prot database, each annotated 84.6% and 48.3% of the 190,755 transcripts, respectively[24,25]. The transcripts were found to match best with genes of other Lamiales species, including *Sesamum indicum*, which covered 75.3% of the transcriptome (Supplementary Fig. 1). 113,221 non-redundant unigenes were categorized by gene ontology (GO) and Kyoto Encyclopedia of Genes and Genomes (KEGG) pathway classification[26,27] (Supplementary Fig. 2). KEGG analysis revealed 5,683 unigenes in 412 KEGG pathways, among which were pathways for terpenoid and steroid biosynthesis (Supplementary Fig. 3). UniProt annotation identified 7517 transcription factors and regulators, 4226 protein kinases, and 22,549 simple sequence repeats (SSR) as genetic markers (Table 1, Supplementary Fig. 4, Supplementary Table 1). The annotated transcriptome presented here provides a comprehensive representation of transcripts in the root and leaf tissues of *D. lanata*.

### Genes for sterol biosynthesis are differentially expressed

Since cardiac glycosides are only present in leaves but not roots (Fig. 1a), we asked if phytosterol and cholesterol biosynthetic genes are overexpressed in leaves. Indeed, genes encoding rate-limiting enzymes in phytosterol and cholesterol pathways were overexpressed in leaves (Fig. 1b). Squalene epoxidase (SQE), a rate-limiting step in sterol biosynthesis, showed higher relative transcript abundance in leaves[28]. Sterol side-chain reductase (SSR1) is a known bottleneck enzyme[29] in cholesterol and phytosterol biosynthesis. Its transcript is also more abundant in *D. lanata* leaves. C4 sterol methyl oxidase 3 (SMO3), unique to the cholesterol pathway, is also more abundant in leaves. It catalyzes the rate-limiting step of 4-methyl elimination in the cholesterol pathway[23]. Indeed, *D. lanata* leaves have higher cholesterol levels than roots, whereas the total sterols in these two tissues are comparable (Supplementary Fig. 5). Another gene with higher transcript abundance in leaves is the sterol C-14 reductase

(C14-R)[30], a shared enzyme between phytosterol and cholesterol pathways.

Analysis of the three known genes involved in digoxin biosynthesis shows that only 3*β*HSD's transcript was more abundant in leaves. While *P5βR*'s relative transcript abundance is the same in both tissues, *P5βR2*'s transcript is more abundant in roots. Since digoxin and sterols are triterpene derivatives, we also analyzed the differential relative transcript abundance of terpenoid biosynthetic genes (Supplementary Fig. 6). The methylerythritol phosphate (MEP) pathway for terpenoid synthesis and triterpene pathways are induced in leaves, agreeing with the compartmentalization of the MEP pathway in the chloroplast[31].

### Identifying two candidate genes as *D. lanata* P450$_{scc}$

The first step of the digoxin pathway is cleaving a sterol by a cytochrome P450$_{scc}$ to generate pregnenolone[13]. Interpro scan identified 438 enzymes annotated as cytochrome P450s (CYPs) (Pfam: PF00067)[32,33] from the transcriptome. CYPs from *Arabidopsis* and CYPs in *D. lanata* were used to construct a phylogenetic tree for CYP subfamily classification (Supplementary Fig. 7). Quantifying relative transcript abundance identified 104 CYP transcripts overexpressed in the leaves (Supplementary Fig. 8). Among these CYPs, only those members of subfamilies relevant to sterol/brassinosteroid biosynthesis were included for future analysis. Thirteen full-length CYP transcripts were identified as potential *P450$_{scc}$* (Fig. 2a). We focused on *DlCYP87A4* and *DlCYP90A1* because *DlCYP87A4* was highly induced in leaves, and CYP90A1 is known to oxidize the 22(S)-hydroxycampesterol[34]. qRT-PCR confirmed that *DlCYP87A4* was expressed much higher in leaves compared to *DlCYP90A1* (Fig. 2b). Therefore, these two transcripts were identified as *P450$_{scc}$* candidates and cloned from cDNA for functional validation by tobacco transient expression assay.

### Tobacco expression identified *D. lanata* P450$_{scc}$ as CYP87A4

To test the two candidates, we employed the tobacco transient expression experiment. Tobacco does not produce digoxin but has sterol substrates for the P450$_{scc}$[35]. Therefore, it is an ideal system for functionally characterizing the P450$_{scc}$ enzyme. The two candidates, *Dl*CYP87A4 and *Dl*CYP90A1, and the two known pathway enzymes, 3*β*HSD and P5*β*R, were expressed in tobacco leaves (Fig. 2c, set 1). Following their expression, products of these enzymes, including progesterone (compound 2), 5*β*-pregnane-3,20-dione (compound 3), and 3*β*-hydroxy-5*β*-pregnane-20-one (compound 4), were detected (Fig. 2c, set 1, Supplementary Fig. 9). The direct product of P450$_{scc}$, pregnenolone (compound 1), was not seen in set 1 potentially due to its quick turnover through 3*β*HSD. Note that the minor peak is not pregnenolone due to the different retention time compared to the pregnenolone standard. In fact, *D. lanata* leaves do not produce detectable amounts of pregnenolone but produce the downstream pathway intermediates (Supplementary Fig. 10). Omitting the *Dl*CYP87A4 abolished the reactions (set 2), whereas taking out the *Dl*CYP90A1 (set 3) had no effect. Expressing *Dl*CYP87A4 alone resulted in the production of pregnenolone (set 4), whereas expressing *Dl*CYP90A1 alone (set 5) did not produce pregnenolone. These data strongly support the hypothesis that *Dl*CYP87A4 is the P450$_{scc}$ of the digoxin pathway.

### Determining the sterol substrates of CYP87A4

The tobacco expression system cannot determine the sterol substrate of *D. lanata* P450$_{scc}$ because tobacco contains a mixture of cholesterol and phytosterols. Thus, we turned to the in vivo yeast expression since yeast does not produce cholesterol or phytosterols. However, feeding yeast with different sterols is challenging due to their hydrophobicity. Therefore, we used previously engineered yeast strains that produce various sterols, including cholesterol, campesterol, 7-dehydrocholesterol, and desmosterol (Fig. 3a, Supplementary Fig. 11)[35]. We also included a wildtype yeast that produces ergosterol. When

## Table 1 | Summary of Transcriptome data

| Transcriptome assembly data | |
| --- | --- |
| Total no. of reads | 173,448,870 |
| No. of reads assembled | 173,445,956 |
| Transcripts | 317,983 |
| N50 value | 1712 bp |
| **Transcriptome Annotation Data** | |
| Full-length transcripts | 121,298 |
| Transcripts annotated by UniProt | 92,133 |
| BUSCO | 94.60% |
| Transcription factors and regulators | 7517 |
| Protein kinases | 4226 |
| simple sequence repeats (EST-SSR) | 22,549 |

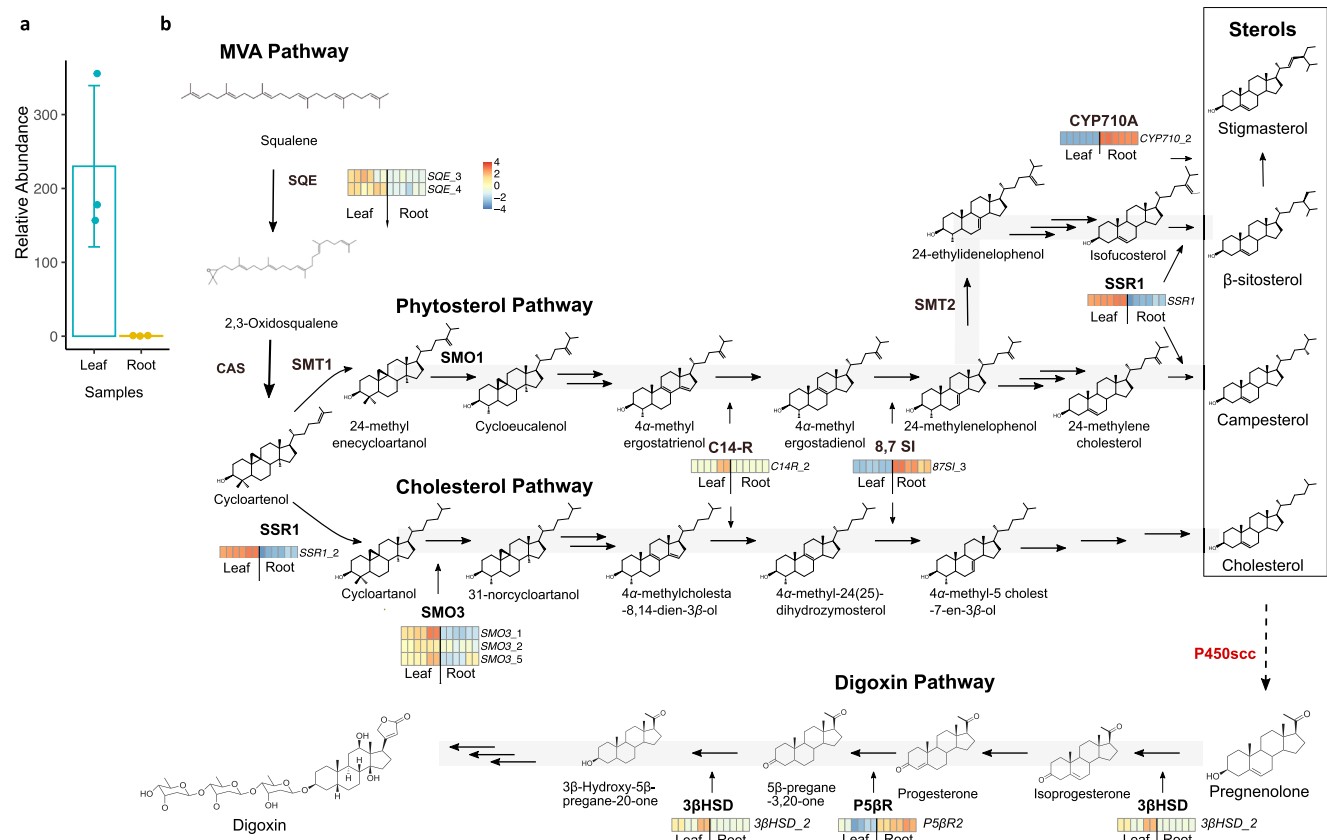

**Fig. 1 | Quantifying the relative transcript abundance of genes in the sterol and digoxin biosynthetic pathways. a** Total lanatosides, including lanatosides A, B, C, and E, in *D. lanata* leaves and roots relative to the deuterated digoxin-d3 internal standard. The data are normalized by dry weight and represent the average ± SD of three biological replicates. **b** Sterol and digoxin biosynthetic pathway genes that are differentially expressed in leaf and root tissues. Genes that do not have

corresponding heat maps are not differentially expressed. SQE squalene epoxidase, CAS cycloartenol synthase, SMO C-4 sterol methyl oxidase, SMT sterol C-24 methyl transferase, SSR1 sterol side-chain reductase 1, C14-R sterol C-14 reductase, 8,7 SI sterol 8,7 isomerase, $P450_{scc}$ cytochrome P450 sterol side-chain cleaving, 3βHSD 3β-hydroxysteroid dehydrogenase, P5βR progesterone-5β-reductase.

expressing the *D. lanata* CYP87A4 and *Arabidopsis thaliana* cytochrome P450 reductase 2 (ATR2) as a redox partner, the yeast strains containing campesterol and cholesterol, respectively, produced pregnenolone (Fig. 3b, Supplementary Fig. 12). Pregnenolone is toxic to yeast; thus, yeast acetylates pregnenolone to detoxify it, generating pregnenolone acetate (compound 5) as a byproduct[36]. We also expressed the human $P450_{scc}$ (CYP11A1) with its redox partners in these yeast strains. Only the cholesterol-producing yeast generated pregnenolone, as expected (Fig. 3b)[18]. These data indicate that both campesterol and cholesterol are substrates of *D. lanata* CYP87A4, which identifies as the *Dl*$P450_{scc}$.

### Neofunctionalization of CYP87A4 unique to *Digitalis*

To understand the evolutionary history of the *D. lanata* CYP87A4, we constructed a phylogenetic tree with transcripts homologous to the *Dl*CYP87A4 from the 1000 transcriptome project (Fig. 4)[37]. Four transcripts in the *D. lanata* transcriptome fall into the CYP87A subfamily. *Dl*CYP87A1 and *Dl*CYP87A2 are 72.2 and 74.2% identical to the characterized *Dl*CYP87A4 (Supplementary Table 2). *Dl*CYP87A3 is 97.4% identical at the protein level to the characterized *Dl*CYP87A4 but does not cleave the side chain of campesterol when expressed in yeast (Supplementary Fig. 14). *Dl*CYP87A1 and *Dl*CYP87A2 likely represent the canonical enzymes of the CYP87A subfamily. Indeed, *Dl*CYP87A1 and *Dl*CYP87A2 are expressed almost constitutively in leaves and roots (Fig. 2a). *Dl*CYP87A4 may be duplicated from a canonical CYP87A and neofunctionalized to gain its sterol cleaving activity. The *Dl*CYP87A4 is distinct from the human $P450_{scc}$ as these two proteins only share 29.8%

identical amino acids (Supplementary Fig. 13). None of the other *Lamiales* in the 1000 transcriptome project had duplicates in the CYP87A subfamily (Fig. 4). Species in the *Oenothera* genus also have multiple copies of CYP87A, but their function is unclear since these species are not known to produce cardiac glycosides.

We included CYP87A transcripts from other plants that produce cardiac glycosides since they would have an enzyme with a similar sterol cleaving function[10,13]. We searched the publicly available transcriptomes of *Digitalis purpurea*, *Calotropis gigantea*, and *Asclepias syriaca* for transcripts that were a close match to the *Dl*CYP87A4. *A. syriaca* did not have any transcripts that matched over 55% at the protein level with *Dl*CYP87A4. *C. gigantea* had one transcript, which matched 69% to *Dl*CYP87A4 (Fig. 4, Supplementary Table 2). The *C. gigantea* CYP87A is likely the canonical CYP87A enzyme since there is only one copy. The *D. purpurea* transcriptome had one transcript that matched 97% to the *Dl*CYP87A4. Thus, this transcript likely has the same sterol cleaving ability. Our analysis indicates that the expansion and neofunctionalization of *Dl*CYP87A as $P450_{scc}$ is probably unique to the *Digitalis* species.

### Identify amino acids critical for *D. lanata* $P450_{scc}$'s function

To gain mechanistic insights into *Dl*CYP87A4's sterol cleaving ability, a protein model was created using AlphaFold2. Campesterol and cholesterol were docked to the active site of the protein model (Fig. 5a, b). Aligning canonical CYP87As and *Dl*CYP87A4's protein sequences identified three unique amino acids in the active site of *Dl*CYP87A4: S123, A355, and L357. These amino acids are conserved between the

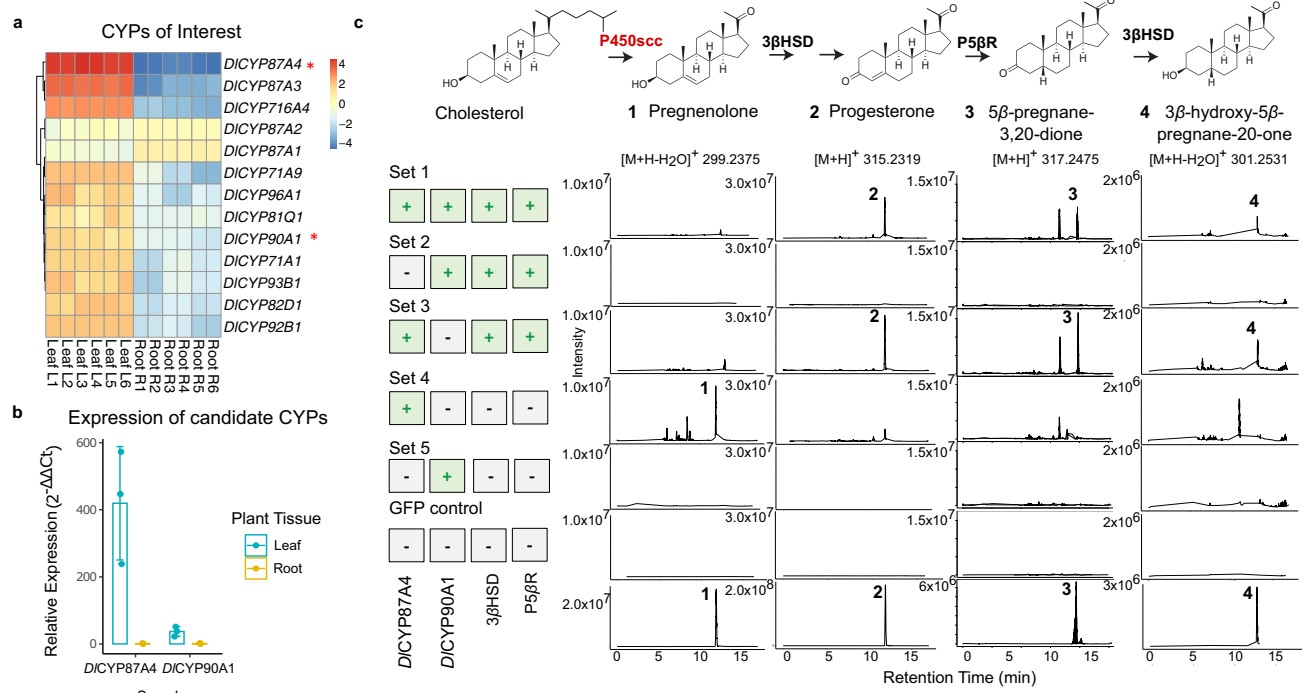

**Fig. 2 | Identifying and characterizing candidate _D. lanata_ P450_scc_s in tobacco.**
**a** Quantifying the relative transcript abundance from candidate genes in root and leaf tissues of _D. lanata_. **b** qRT-PCR quantified relative transcript abundance of top candidate genes, _Dl_CYP87A4 and _Dl_CYP90A1. Data represent the average ± SD of three biological replicates. **c** LC/MS data from tobacco leaves transiently expressing candidate genes in various combinations. The theoretical m/z values of parent ion adducts are given. Detected m/z values of peaks are within 5 ppm of the theoretical value. Set 1 contained full-length _Dl_CYP87A4, _Dl_CYP90A1, 3βHSD, and P5βR. Note the minor peak is not pregnenolone because of the different retention time compared to the pregnenolone standard. Set 2 contained _Dl_CYP90A1, 3βHSD, and P5βR. Set 3 contained _Dl_CYP87A4, 3βHSD, and P5βR. Set 4 contained _Dl_CYP87A4 only. Set 5 contained _Dl_CYP90A1 only. GFP control: negative control expressing a green fluorescent protein (GFP) in tobacco. The bottom panel shows the authentic standards of the four expected pathway intermediates.

_Dl_CYP87A4 and the putative _D. purpurea_ P450_scc_ (_Dp_CYP87A) but differ from the canonical CYP87As (Fig. 5c), suggesting that they are important for the sterol cleaving activity. Reverting A355 to leucine or L357 to alanine, as in the canonical _Dl_CYP87A1, abolished the campesterol side chain-cleaving activity, whereas the S123A mutation had no effect (Fig. 5d). A355 and L357 likely stabilize the steroid by forming hydrophobic interactions with the four steroid rings (Fig. 5a, b). However, these two amino acids are insufficient to impart the sterol side-chain-cleaving activity as the canonical _Dl_CYP87A1 mutated with these two amino acids was unable to cleave campesterol (Supplementary Fig. 14). The wildtype canonical _Dl_CYP87A1 could not cleave campesterol, suggesting it is not involved in digoxin biosynthesis (Fig. 5e).

## Discussion

We identified and characterized the first and rate-limiting enzyme in the plant cardiac glycoside biosynthetic pathway, P450_scc_, which has long been speculated but not found before. We used differential transcriptomic analysis to identify that a CYP87A family protein acts as the P450_scc_ in _Digitalis_. This protein is distinct in sequence from its mammalian counterpart, CYP11A1. The similarities and differences between the mammalian and plant P450_scc_ indicate that the "cholesterol side-chain-cleaving" activity evolved independently and serves distinct functions. While mammalian P450_scc_s for steroid hormone biosynthesis are essential for the normal development of animals, plant P450_scc_s evolved for synthesizing specialized metabolites, such as cardiac glycosides, which are unique to very specific plant families.

The _Dl_P450_scc_ is a crucial "gatekeeping" enzyme that connects plant primary and secondary metabolisms. It channels sterols essential for maintaining cell membrane homeostasis to produce cardiac glycosides, secondary metabolites important for defense[23]. Such an enzyme acting as the "gatekeeper" for specialized metabolism is not surprising. The rate-limiting nature of _Dl_P450_scc_ was evident because feeding cholesterol to _Digitalis_ produced a trace amount of pregnenolone, whereas administering progesterone increased various pregnane intermediates[17]. Unlike P450_scc_ in animals, _Dl_P450_scc_ is promiscuous as it catalyzes the side-chain-cleaving reaction for cholesterol and campesterol (Fig. 3). This promiscuity is somewhat expected since campesterol is one of the major sterols in plants. In order to test if the other major plant sterols, β-sitosterol and stigmasterol, may also serve as substrates for _Dl_P450_scc_, we performed a docking simulation (Fig. 6). While the C20 and C22 of campesterol, cholesterol, and β-sitosterol are within 4.6–5.6 Å to the heme center, docking with stigmasterol put these two carbons over 7 Å away from the heme. The 22:23 double bond of stigmasterol prevents bond rotation resulting in the bulky 22-ethyl group pointing towards the heme center, preventing C20 and C22 from accessing the heme. Thus, we surmise that β-sitosterol could also be _Dl_P450scc's substrate, along with cholesterol and campesterol, but not stigmasterol. The discovery of _Dl_P450_scc_ as a promiscuous protein will likely end the half-a-century controversy over the sterol precursor for digoxin.

Future work is necessary to understand if the _Dl_P450_scc_ acts by the exact catalytic mechanism as the mammalian P450_scc_, which catalyzes three-step sequential oxidations through 22-hydroxylation, 20-hydroxylation, and cleavage between C20 and C22[18]. Previous in vitro assay using 20- or 22-hydroxycholesterol as substrates support this mechanism[17]. We show that A355 and L357 are essential for _Dl_P450_scc_'s activity (Fig. 5d). These two amino acids are unique to _Dl_P450_scc_ and distinct from canonical CYP87As (Fig. 5c). They likely form a conformationally optimized "floor" ideal for binding sterols in the enzyme's active site (Fig. 5a, b). However, since the activity assay used cell lysate instead of purified proteins, which are difficult to isolate

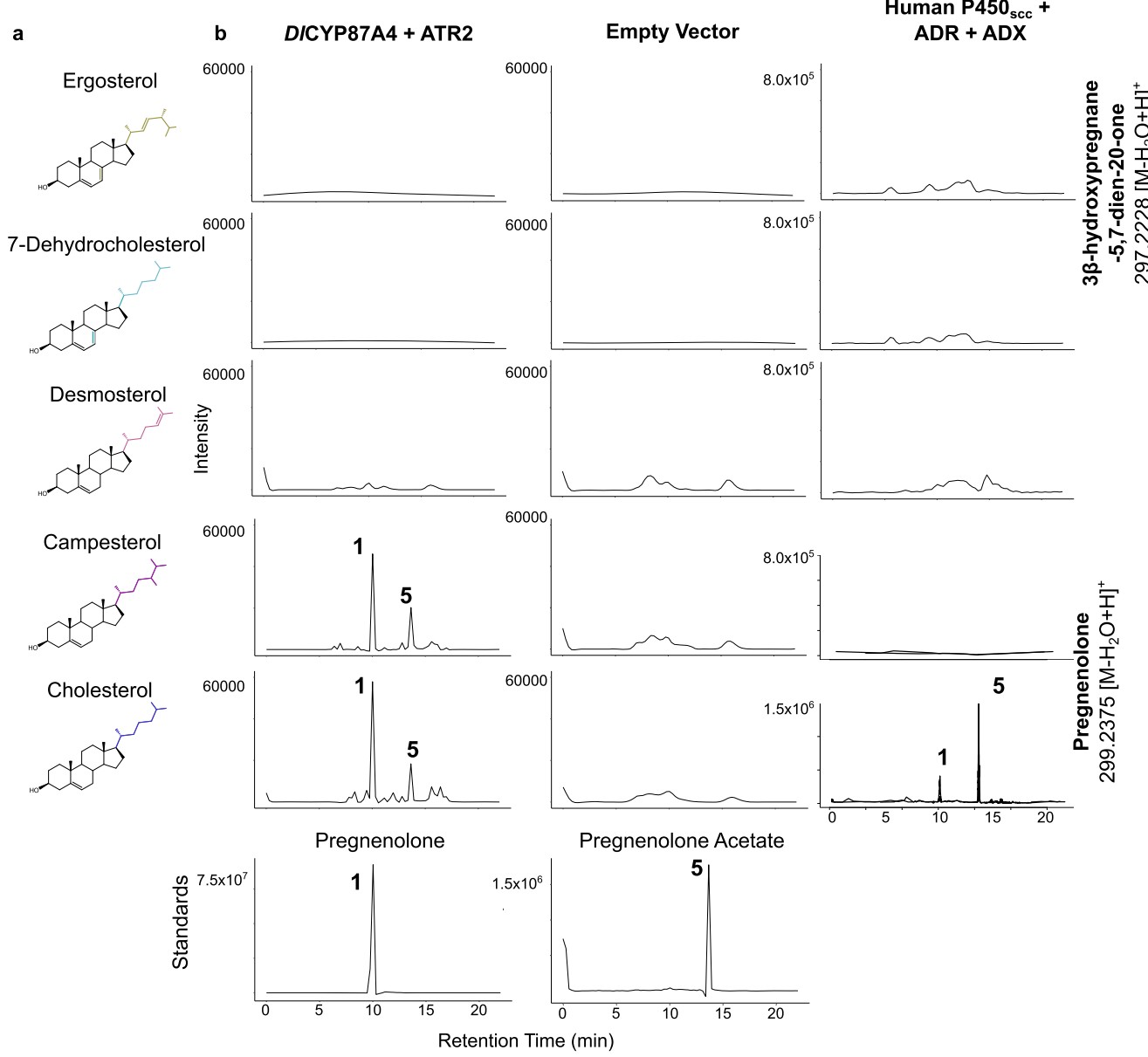

**Fig. 3 | Characterizing the *D. lanata* CYP87A4 in *S. cerevisiae* strains producing various sterols. a** Sterols the yeast strains produce. **b** Extracted ion chromatograms from yeast expressing the full-length *Dl*CYP87A4 and ATR2 or the human P450scc with its redox partners, adrenodoxin (ADX) and adrenodoxin reductase (ADR). The theoretical m/z values of parent ion adducts are given. Detected m/z values of peaks are within 5 ppm of the theoretical value. The bottom panel shows authentic standards of pregnenolone and pregnenolone acetate.

despite repetitive attempts, we do not rule out the possibility that these two mutations may also affect protein folding or stability. Comparing the substrate recognition sites of *Dl*P450scc and the human P450scc revealed that the amino acids are distinct, although both sites are comprised mainly of non-polar amino acids, indicating hydrophobic interactions are the main driving force for substrate binding (Fig. 7). It remains unclear, however, if the stereochemistry of the 24-methyl group of campesterol affects the catalytic activity of *Dl*P450scc. Many plant species contain an epimeric mixture of 24(R)- and 24(S)-campesterol[38–40]; the latter is called dihydrobrassicasterol. It is unclear if *D. lanata* contains both epimers or only the 24(S) stereoisomer.

The identification of *Dl*P450scc will enable the study of cardiac glycoside biosynthesis in other plant species, such as milkweed (*Asclepias, Calotropis*), wallflower (*Erysimum*), and oleander (*Nerium oleander*), to name a few[41]. Phylogenetic analysis showed that *Calotropis gigantea* might not have a duplicated CYP87A gene (Fig. 4), assuming the publicly available *Calotropis* transcriptome is complete. Interestingly, the CYP87A is in the same phylogenetic clade as CYP90B1 that catalyzes the 22(S)-hydroxylation of campesterol, which is one of the three steps in the sterol side-chain cleaving reaction (Supplementary Fig. 7)[18]. It is likely that cytochrome P450s within this clan, including CYP708A, CYP88A, CYP702A, CYP85A, CYP90, CYP720A, and CYP724A, have the potential to evolve the sterol side-chain-cleaving activity. It remains unclear what is the function of the canonical CYP87A. It may oxidize a sterol or a triterpenoid since CYP87D16 from *Maesa lanceolata* oxidizes the C16 of β-amyrin[42].

In conclusion, this work identified the rate-limiting and long-speculated P450scc in *Digitalis* for the biosynthesis of digoxin. It is an essential step toward ultimately elucidating the digoxin biosynthetic pathway. This work will also open the door for biomanufacturing novel digoxin analogs with expanded medicinal value in microbial or plant systems.

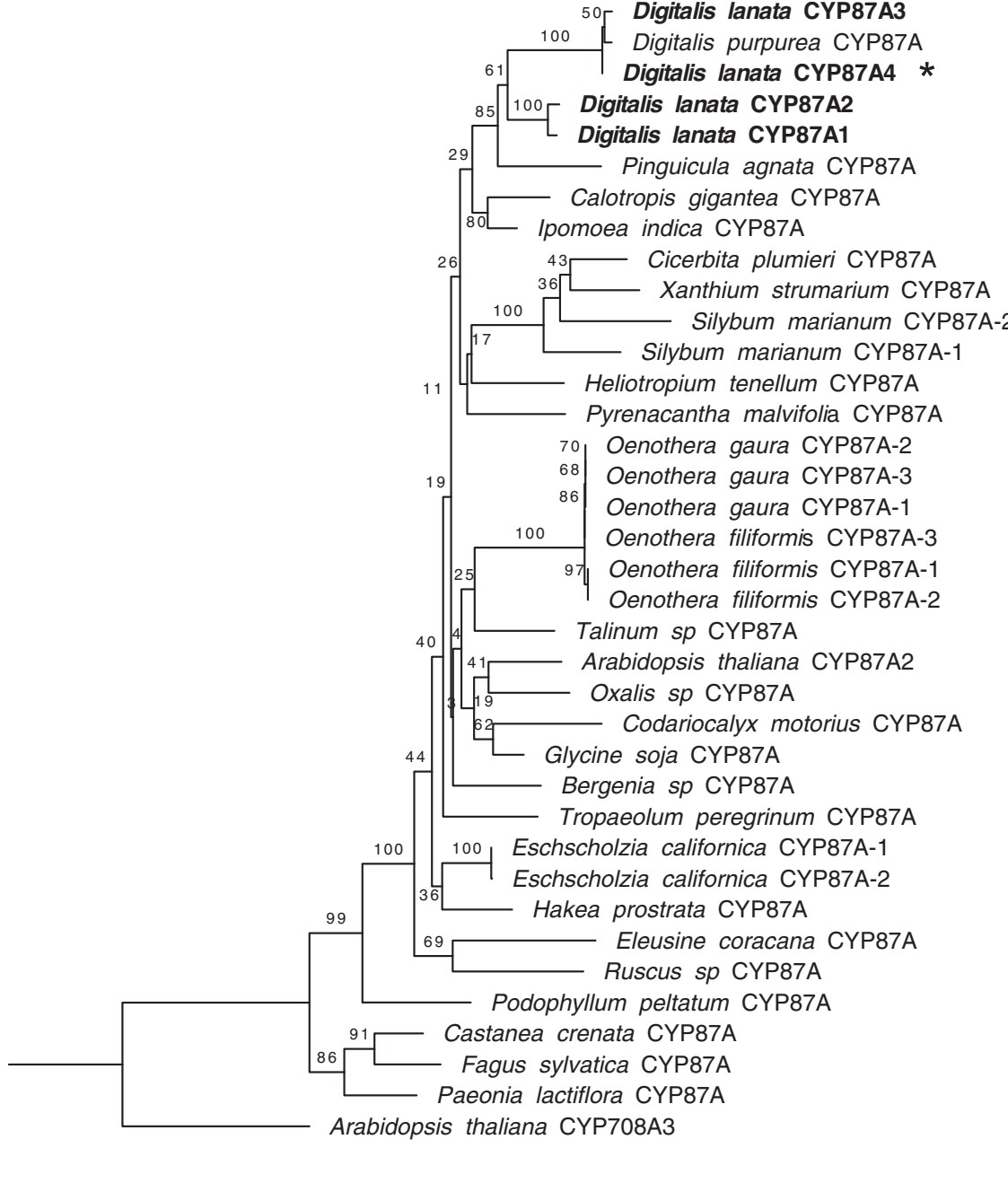

**Fig. 4 | Maximum-likelihood phylogenetic tree of the CYP87A family members from eudicot plants.** Protein sequences are retrieved from 1000 plant transcriptome project and are at least 50% identical to *Dl*CYP87A4. Bootstrap values from 1000 replicates are shown at each node. *D. lanata* CYP87As are bolded. Asterisk denotes the characterized *Dl*CYP87A4 in this study. The scale bar represents the mean number of substitutions per amino acid.

## Methods

### Plant material, RNA isolation, and sequencing

*Digitalis lanata* Ehrh seeds were procured from Strictly Medicinal (Williams, Oregon, USA). Seeds were germinated on the soil mix (57 g triple superphosphate, 85 g calcium hydroxide, 57 g bone meal, 369 g Osmocote (14-14-14), 99 g calcium carbonate, 25 L perlite, 50 L loosened peat and 25 L coarse vermiculite) and maintained in a growth chamber (Invitrogen, Clayton, Missouri, USA) under a light period of 16-h at 25 °C and a relative humidity of 60–80%.

Leaf and root tissues from three different seedlings were used to prepare the Illumina sequence library. Each seedling represents one biological replicate, and the total RNA from each replicate is split into two technical replicates. Total RNA was isolated using the RNeasy Plant Mini Kit (Qiagen, Germantown, MD, USA). The sequencing library was prepared from total RNA using the TrueSeq Ribo-Zero Plant RNA library prep kit (Illumina, San Diego, CA, USA) that removes ribosomal RNA. A quality check of the library was carried out with an Agilent 2100 bioanalyzer. The library was sequenced using Illumina HiSeq 2500 to generate 100 bp paired-end raw reads.

### Gene isolation and cloning

The detailed cloning method is included in the Supplementary Methods. Primers used are listed in Supplementary Table 3, and plasmid constructs are listed in Supplementary Table 4, respectively.

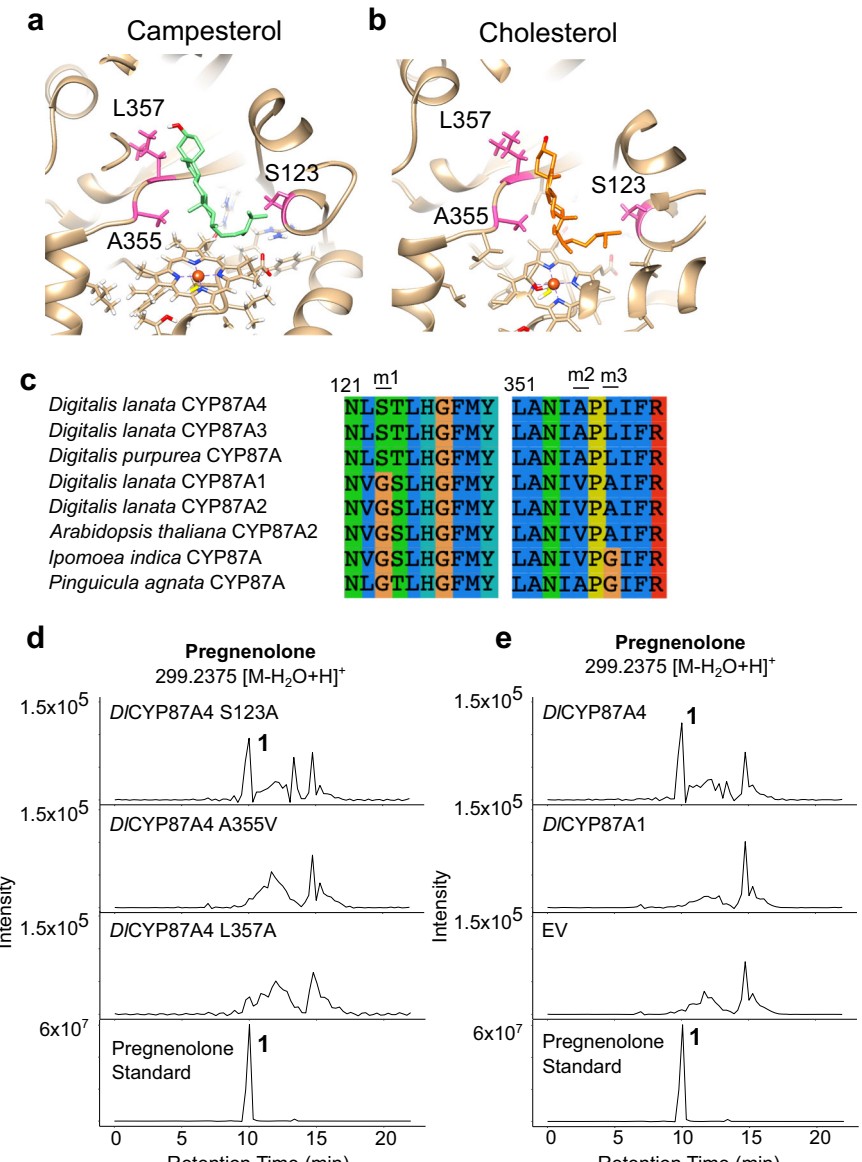

**Fig. 5 | Protein modeling identified critical amino acids for *Dl*CYP87A4's sterol cleaving activity.** Docking of campesterol **a** and cholesterol **b** into the active site of the *Dl*CYP87A4 protein model. Side chains of key amino acids S123, A355, and L357 are highlighted in pink. The protein model is truncated of a predicted N-terminal signal peptide of 31 amino acids. **c** Alignment of CYP87A subfamily enzymes to *D. lanata* and other CYP87A enzymes. Three locations that are different between *Dl*CYP87A4 and the canonical *Dl*CYP87A1 are indicated as follows; m1 (S123), m2 (A355), and m3 (L357). **d** Extracted ion chromatograms from campesterol-producing yeast expressing *Dl*CYP87A4 mutants, including S123A, A355V, and L357A, along with ATR2. The theoretical m/z values of parent ion adducts are given. Detected m/z values of peaks are within 5 ppm of the theoretical value. **e** Campesterol-producing yeast expressing the wildtype *Dl*CYP87A4 or the canonical *Dl*CYP87A1 along with ATR2. Authentic standard for pregnenolone is shown in the bottom panel.

## Tobacco transient expression

**Agrobacterium transformation.** pEAQ plasmids carrying genes of interest were transformed into the *Agrobacterium tumefaciens* strain AGL1 individually by the freeze-thaw method[43]. The resulting strains were prepared for infiltration using a modified protocol as in Saxena et al.[44]. Briefly, A single *Agrobacterium* colony containing one of the pEAQ plasmids was inoculated into 5 ml yeast extract broth (YEB) [5 g/L tryptone, 1 g/L yeast extract, 2.5 g/L Luria broth (Fisher Scientific, Waltham, MA, USA), 5 g/L sucrose, 0.49 g/L MgSO$_4$·7H$_2$O] with 50 mg/L kanamycin for pEAQ plasmid selection and 25 mg/L rifampicin for *A. tumefaciens* strain AGL1 selection. The bacterial cultures were grown 24 h at 28 °C with shaking at 220 rpm. Afterward, 0.5 ml of the seed culture was used to inoculate 25 ml of YEB with kanamycin (50 mg/L) and rifampicin (25 mg/L). The flasks were grown overnight at 28 °C, 220 rpm. The cultures were pelleted at 3000 g for 15 min, washed once with 10 mL sterile double-distilled water (ddH$_2$O), and resuspended in MMA [10 mM MES (2-N-morpholinoethanesulfonic acid), pH 5.6, 10 mM MgCl$_2$, 100 μM acetosyringone]. The individually transformed strains were pooled together so that the final volume was 10 ml and each *A. tumefaciens* strain had a final OD$_{600}$ of 0.4. Then cultures were incubated for 2 to 4 h at 28 °C before infiltrating tobacco leaves.

**Tobacco infiltration.** The pooled *A. tumefaciens* was infiltrated into the underside of four- to six-week-old *Nicotiana benthamiana* new leaves using a needleless plastic syringe. The tobacco plants were grown in 16-h light and 8-h dark periods at 21 °C with a relative humidity of 60–80% and photon intensity of 120–150 μmol/m². Three leaves were infiltrated for each experimental set, and each set was completed on a single plant. As a negative control, *A. tumefaciens* transformed with pEAQ_*GFP* was infiltrated into a separate plant. Plants

were maintained in dark for 12 h to increase the agrobacterial infection and then shifted to light. The plants were maintained in normal conditions for four to six days. Once the fluorescence from GFP was intense when exposed to UV light, all infiltrated leaves were detached from the petiole, snap-frozen in liquid nitrogen, and ground into a fine powder. Metabolites were extracted with 1 ml 100% methanol (Fisher Scientific, Waltham, MA, USA) and heating at 65 °C for 10 min. They

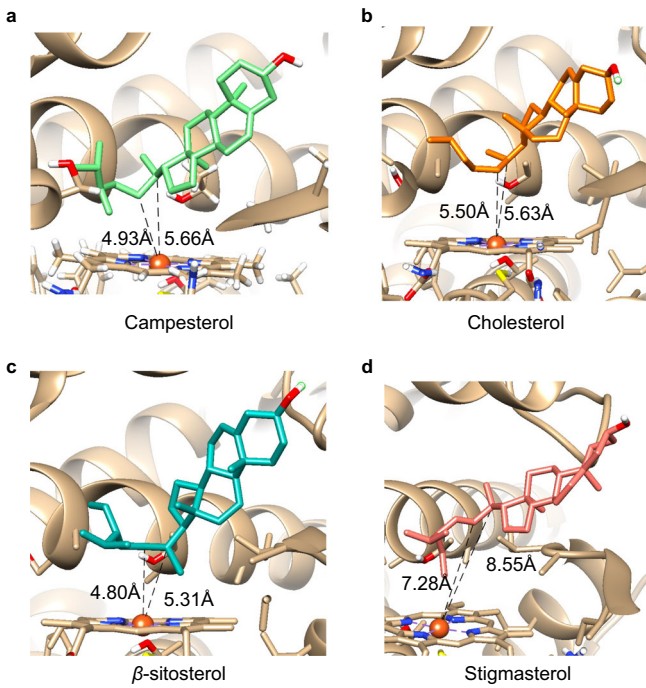

**Fig. 6 | Docking cholesterol and phytosterols to the active center of the *Dl*P450scc.** Docking simulations of *Dl*CPY87A4 with **a** campesterol, **b** cholesterol, **c** β-sitosterol, and **d** stigmasterol. Dashed lines show the distances of C20 and C22 to the heme center. Docking conformations shown are of the lowest energy in each simulation.

were centrifuged at 17,000 g for 10 min, filtered through a 0.45 µm filter (VWR, Randor, PA, USA), and stored at −20 °C before LC/MS analysis.

### Yeast in vivo expression assay

Sterol-producing yeasts were kindly provided from the Riezman lab (Supplementary Table 5)[35]. Competent cells of these strains were prepared using the Frozen EZ Yeast Transformation II Kit™ (Zymo Research, Irvine, CA, USA). Starter cultures were grown in the SD-Leu medium at 30 °C overnight and then used to inoculate 25 mL SD-leu medium in triplicates in a shaking flask with an initial $OD_{600}$ of 0.2−0.4. Samples were harvested at 18 h and pelleted 3000 g for 5 min. Yeast cells were resuspended in 200 µL of TES buffer (50 mM Tris-HCl pH = 7, 600 mM sorbitol, 10 g/L bovine serum albumin, 1.5 mM β-mercaptoethanol) and homogenized with an equal volume of 0.5 mm glass beads in a BBX24 Bullet Blender® homogenizer (Next Advance, Troy, NY, USA) at setting 8 at 4 °C for 4 min. A total of 300 µL of TES buffer was added to the lysed cells, and 400−500 µL of the yeast lysate was transferred into a capped glass tube, followed by adding 1 mL chloroform immediately. The sample was vortexed for 1 min, and the organic phase was transferred into a new glass test tube and dried under a stream of air. The sample was resuspended in 100 µL methanol, centrifuged at 17,000 g for 10 min, and the supernatant was transferred into a LC/MS vial and stored at −20 °C until use.

### LC/MS analysis for pregnane intermediates in the digoxin pathway

Samples were analyzed using a LC/MS² instrument, a Thermo Scientific Q-Exactive Focus™, a hybrid quadrupole and orbitrap mass analyzer (Fisher Scientific, Waltham, MA, USA) and Thermo Scientific UltiMate 3000 UHPLC™ (Fisher Scientific, Waltham, MA, USA). A Waters XSelect CSH™ C18 HPLC column (SKU: 186005257, Waters, Milford, MA, USA) with a particle size of 3.5 µm, an internal diameter of 2.1 mm, and a length of 150 mm was used for separation. The column was set to 25 °C with the back pressure in the range of 130−150 psi. *Digitalis lanata* and tobacco extract samples in biological triplicates were analyzed as previously described[45,46]. For analyzing pregnenolone and pregnenolone acetate from yeast samples, the following protocol was

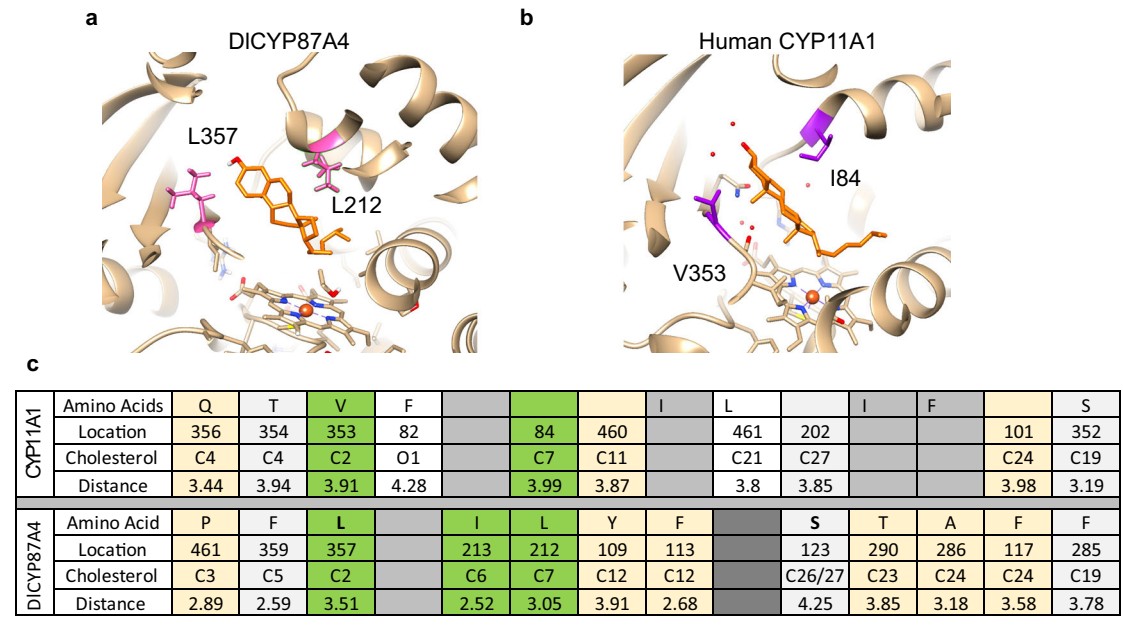

**CYP11A1**

| | Amino Acids | Q | T | V | F | | | I | L | | I | F | | S |
|---|---|---|---|---|---|---|---|---|---|---|---|---|---|---|
| | Location | 356 | 354 | 353 | 82 | | 84 | 460 | | 461 | 202 | | 101 | 352 |
| | Cholesterol | C4 | C4 | C2 | O1 | | C7 | C11 | | C21 | C27 | | C24 | C19 |
| | Distance | 3.44 | 3.94 | 3.91 | 4.28 | | 3.99 | 3.87 | | 3.8 | 3.85 | | 3.98 | 3.19 |

**DlCYP87A4**

| | Amino Acid | P | F | L | | I | L | Y | F | | S | T | A | F | F |
|---|---|---|---|---|---|---|---|---|---|---|---|---|---|---|---|
| | Location | 461 | 359 | 357 | | 213 | 212 | 109 | 113 | | 123 | 290 | 286 | 117 | 285 |
| | Cholesterol | C3 | C5 | C2 | | C6 | C7 | C12 | C12 | | C26/27 | C23 | C24 | C24 | C19 |
| | Distance | 2.89 | 2.59 | 3.51 | | 2.52 | 3.05 | 3.91 | 2.68 | | 4.25 | 3.85 | 3.18 | 3.58 | 3.78 |

**Fig. 7 | Substrate binding sites of the *Dl*P450scc and the human P450scc.** Comparison of substrate recognition sites of *Digitalis lanata* CYP87A4 **a** and human CYP11A1 **b**. Cholesterol is docked to the active sites. Amino acids at near-identical positions of *Dl*CYP87A4 and CYP11A1 are shown. **c** List of active-site amino acids within 4.5 Å to cholesterol in *Dl*CYP87A4 and human CYP11A1. Green: similar amino acids in the two active sites; tan: hydrophobic amino acids in both active sites; gray: distinct amino acids.

developed. The mobile phase A was water with 0.1% formic acid and mobile phase B was acetonitrile with 0.1% formic acid, with a flow rate of 200 μL min⁻¹. Gradient started with 40% mobile phase B for 2 min followed by a linear gradient of 40% to 95% B from 2 to 11 min, held for 5 min, and brought back to initial conditions of 40% mobile phase B in 6 min. The sample injection volume was 20 μL, and the injection cycle time set to automatic with no sample splitting. The eluents were ionized by electron spray ionization (ESI) and analyzed in the positive ion mode. The full scan range is 100–1200 *m/z* at resolution of 70,000, inclusion error of ±5 ppm and automatic gain control (AGC) of 1 million. The resolution for ddMS$^2$ is 17,500 with collision energies at 10, 30, and 60 eV, isolation window of 3.0 m/z, and AGC of 8000. The scan rate was set at automatic. The mass spectrometer was regularly calibrated to ensure mass accuracy. Qualitative analysis was performed using the XCalibur™ (v. 4.4.16.14) software. Raw files were converted to.mzML files using MSConvert (v. 3.0.21040), and chromatograms and spectra were generated in R (v. 4.2.0) using the XCMS (v. 3.18.0) and Spectra (v. 1.6.0) packages[47,48].

### Phylogenetic analysis

Transcriptome sequences used for the CYP87A tree were retrieved by BLASTing the *Dl*CPY87A4 transcript against the 1000 Plant Transcriptome (1KP) database using tBLASTx[37,49] under default parameters. Sequences were filtered, and only those within 400–600 amino acids long and had a start codon were retained. *A. thaliana* CYP87A2 was used as a reference, and CYP708 was used as an outgroup.

All trees were constructed by aligning protein sequences using MAFFT[50]. Aligned sequences were trimmed using trimAl[51]. The phylogenetic tree was constructed using RAxML-NG (v. 1.0.1) with the all-in-one Maximum likelihood (ML) tree search and slow bootstrapping with 1000 replicates[52].

### Protein modeling and docking

A protein model for the *D. lanata* P450$_{scc}$ (*Dl*CYP87A4) was generated using Alphafold2 through the ColabFold platform v1.4[53,54]. The MSA mode used was MMseqs2, and all other parameters were the default[55]. Five models were generated, and model 3 was used for further analysis based on Predicted Aligned Error (PAE) and predicted local distance difference test (pLDDT) scores (Supplementary Fig. 15). Docking of sterols was performed using Chimera version 1.16 and Autodock Vina version 1.1.2[56,57].

### Statistics

**Total lanatoside quantification.** The data are normalized by dry weight and represent the average ± SD of three biological replicates.

**Transcriptome heatmaps.** Three biological replicates with two technical replicates each from roots and leaves were used to generate heatmaps. EdgeR was used to conduct the differential expression analysis to obtain log2FPKM (transcript per million mapped reads). The significance cutoff for overexpression in leave is $P < 0.05$.

**qRT-PCR.** Three biological replicates and two technical replicates were included for each sample. The mean of the two technical replicates' Ct values was normalized against that of the polyubiquitin 10 gene (*UBQ10*) to calculate the ΔCt. ΔCt value was then normalized against the mean ΔCt of roots to derive the ΔΔCt value.

### Reporting summary

Further information on research design is available in the Nature Portfolio Reporting Summary linked to this article.

## Data availability

Data generated and analyzed are included in the published article and its supporting information files. *D. lanata* raw RNA-seq reads and the assembled transcriptome are deposited into the Gene Expression Omnibus database (Accession: GSE224014). LC/MS$^2$ and GC/MS data are available in the Metabolites database (Accession: MTBLS7993). *D. lanata* CYP87A1-4 sequences are available in Supplementary Data 1, 2 and GenBank database (Accession: OR134561, OR134562, OR134563, OR134564). Source data are provided with this paper.

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

## Acknowledgements

We thank Dr. Howard Riezman at the University of Geneva for providing the sterol-producing yeast strains, Rian Hammond for providing plasmids encoding the human P450scc, Dr. George Lomonossoff at the John Innes Center and Leaf Systems for supplying the pEAQ vector, Dr. Valerie Freichs for assistance with chromatography work, and Dr. Donald Yergeau for RNA-seq at University at Buffalo. This project was supported by the Research Foundation for the State University of New York [71272] to Z. Q. Wang and the National Science Foundation [CHE-1919594] to the University at Buffalo Chemistry Instrument Center.

## Author contributions

E.C., B.R.G., and Z.Q.W. designed research; E.C., B.R.G., I.R. and M.M. carried out experiments; E.C., B.R.G, I.R., M.M. and Z.Q.W. analyzed data; E.C., B.R.G, I.R. and Z.Q.W. wrote the paper.

## Competing interests

The authors declare no competing interests.
