## [Peer Review File · Nature Communications]

A Cytochrome P450 CYP87A4 Imparts Sterol Side-Chain Cleavage in Digoxin BiosynthesisReviewer #1 (Remarks to the Author):

The article by Carroll et al. reports on the discovery of a key enzyme in cardenolide biosynthesis. That enzyme is the elusive sterol side chain cleavage enzyme. This essential nature of this enzyme activity has been recognized since the first predictions for digoxin biosynthesis many decades ago. The enigmatic and elusive nature of this enzyme resided in the suspicion that it would resemble the sterol side chain cleavage enzyme prevalent in mammals and essential for steroid hormone biosynthesis. What the current report demonstrates is that a somewhat less biased approach was necessary to discover the gene encoding for this enzyme activity.

The authors firstly developed a comprehensive transcriptome for *Digitalis lanata*, then compared the transcript levels for those known enzymes in the digitoxin biosynthetic pathway in leaves and roots. Interestingly, while it is well appreciated that the cardenolides accumulate in leaves, only two of the previously characterized digitoxin biosynthetic genes showed a slight elevated transcript level in leaves. When the transcriptome was probed for CYP mRNAs because this is the class of enzymes associated with sterol side chain cleavage in mammals, several members of the CYP87 mRNA family were strongly elevated in leaf tissue. This led to a functional characterization of one particular member, CYP87A4. When transiently expressed in tobacco, pregnenolone was detected, and when co-expressed with other digitoxin biosynthetic genes other downstream cardenolide intermediates were documented. When these *in vivo* functional tests were run in yeast engineered for accumulating various sterols, it was demonstrated that campesterol, like cholesterol, could also serve as a direct precursor for pregnenolone. This was contrasted with direct evidence that the mammalian sterol cleavage enzyme demonstrated selectivity for only cholesterol. This in turn focused the authors' attention on examining the catalytic residues within the CYP87A4 for their role in catalysis and substrate selectivity. Unfortunately, they only observed knockout mutants for use of campesterol and thus did not obtain information about how the CYP87A4 enzyme imposed differential substrate selectivity in comparison to the mammalian sterol cleavage enzyme.

Overall, the work presented in this manuscript is well done and the identification of the essential sterol side chain cleavage enzyme is compelling. What isn't so well done is the written text. Firstly, the manuscript itself should be written without necessity to consult the supplemental information about the experiments. For instance, what are the 6 treatment samples for leaf and root in the transcript profiling experiments? Duplicates of the biological replicates? Why weren't leaf and root tissue differentiated by age to give a dynamic associated with digitoxin accumulation? A small item, but nonetheless illustrative of the writing style, the transcript profiling is about relative transcript abundance, not expression. In Figures 2 and 3, I'm very interested in how much of the available campesterol and cholesterol are converted to products. This is a measure of relative enzyme efficiencies and important to distinguish between a true enzymatic conversion versus a promiscuous activity. The text description of Fig. 2 also suggests that pregnenolone isn't observed, but the actual figure suggests otherwise. A lot of these issues could be addressed by inclusion of quantitative assessments (how much substrate converted to product) and careful attention to the writing style.

While the data is compelling, my greatest concern is that the authors have not crafted the document addressing a more significant issue of greater interest to a general readership. This relates in part to their phylogenetic inspection. They really do not address the issue that plants have evolved a sterol side chain cleavage enzyme unlike that in mammals. The upshot of this is these two kingdoms have evolved two very different classes of steroidal based hormones, one requiring a SCC cleavage enzyme, the other not. But that then provided a means for plants to evolve a SCC cleavage enzyme for yet another function, for cardenolide biosynthesis. And this is unique to very specific plant families.

Reviewer #2 (Remarks to the Author):

The study identified an unknown P450 catalyzing cholesterol side-chain cleavage in digoxin biosynthetic pathway, which contributes most to decode the complete digoxin biosynthetic pathway in foxglove. The P450 was screened by differential transcriptomic analysis, and further verified by both tobacco and yeast expression experiments. From the point of gene discovery, the gene was not the first identified gene with the function and the digoxin biosynthetic pathway had been reported previously. Overall, the research does not contain sufficient novelty and new findings for publishing in NC.

Some minor suggestions:

1. Given that the industrial significance of the catalytic function of sterol side chain cleaving, it's equally important to get a higher activity of enzyme. We hope the author can compare the catalytic activity of DICYP87A4, DICYP87A4's homologs and other reported P450scc.
2. The study revealed two amino acids which is critical for CYP87A's catalyzation. We hope the author can discuss the similarities and differences of key residues in catalytic pocket between CYP87A and CYP11A1.
3. Line 57. CYP78A or CYP87A ?
4. Line 128. Therefore, these two transcripts.., these ?

Reviewer #3 (Remarks to the Author):

This manuscript describes investigation of the biosynthesis of digoxin, a truncated (hexa-nor) triterpenoid saponin from foxglove (*Digitalis lanata*) widely used for treating congestive heart failure. Despite its clear medicinal relevance, relatively little has been previously elucidated about digoxin biosynthesis, with only a dehydrogenase and reductase identified. Here a high-quality transcriptome was generated from roots and leaves, and used to identify the relevant cytochrome P450 (CYP) catalyzing side-chain cleavage, which initiates digoxin biosynthesis. The work is solid and quite well-described. Given the association of digoxins with leaves, and up-regulation of at least the known dehydrogenase and certain up-stream acting enzymes, two CYPs were selected for investigation on the basis of such up-regulation as well as known activity for other (sub)family members in phytosterol/brassinosteroid biosynthesis – DICYP87A4 and DICYP90A1. DICYP87A4 was found to catalyze side-chain cleavage in recombinant settings, with the use of yeast with modified triterpenoid metabolism enabling investigation of substrate specificity and revealing activity with either the expected substrate cholesterol or campesterol, which is more commonly found in plants. Further phylogenetic analysis suggests that expansion of the CYP87A sub-family likely underlies the presence of such triterpenoid truncation for cardiac glycoside biosynthesis in the *Digitalis* genus, but not in other plant lineages. Finally, protein modeling, with substrate docking and comparison of these 'extra' CYP87A sub-family members from *Digitalis* relative to the generally single copies more widely found, identified two residues whose change in identity is important for the sterol side-chain cleavage activity. Unfortunately, it remains unclear what role the ancestral CYP87A sub-family members play in plant metabolism, so only loss of activity could be determined here. On the other hand, it might be of interest to determine if these two changes are sufficient to impart the ability to catalyze sterol side-chain cleavage. Regardless, while the known ability of other CYPs (e.g., that from humans noted here) to catalyze at least the conversion of cholesterol to pregnenolone (i.e., sterol side-chain cleavage) reduces the novelty of this finding, the data reported here at the very least provide key insight into this important metabolic process, and an ideal foundation for further studies as well.

Minor corrections:

Ln 40: "works" should be "work"

Ln 64: "performed the de novo" should be "performed de novo"

Ln 221: "5.66 Å 4.93" should be "5.66 Å and 4.93"

Reviewer #4 (Remarks to the Author):

This manuscript describes the discovery and characterization of a foxglove CYP450 enzyme key to digitoxigenin biosynthesis. The candidate P450-encoding genes were identified from transcriptome

assembly based on differential expression analysis, and via heterologous expression followed by in vivo assays in both tobacco and yeast systems, the leaf-abundant DICYP87A4(?) was shown to serve as a P450_{sc} cleaving side chain of sterol substrates. Then phylogenetic analyses highlighted a possibility that *Digitalis* CYP87As might undergo duplication and neofunctionalization for more specialized metabolism. Key residues affecting side chain-cleaving activity were also investigated. Characterization of the first P450_{sc} in plant is an important step for deciphering complete pathway of cardiac glycosides. Experiments of this paper are generally well conducted. However, I think this is a routine story accomplished by common strategies starting from mining of RNA-seq data to functional identification. Only one p450 gene has been reported, which does not represent enough novelty suitable for acceptance.

Additional comments:

1. In Fig 1, please enlarge the chemical structures. The structure of isoprogesterone is not correct.
2. I wonder if beta-sitosterol and stigmasterol can be served as the substrates of DICYP87A4, since these sterols are more generally present in plants.
3. It is not clear why the cloning of CYP87A3 failed. Can this ORF be synthesized for test.
4. What are the parameters of tBLASTx search for CYP87A transcripts? Line 207, "canonical CYP87A clade", which branches in the tree have been classified as canonical group? The ancestral function of CYP87A (presuming more conserved function involved in primary metabolism) is currently unknown. This prevents author's interpretation of the evolutionary path of specialized CYP87A.

Response to reviewers

The authors would like to thank the reviewers for the thoughtful comments and helpful suggestions, which have been incorporated into the revised manuscript "A Cytochrome P450 CYP87A Imparts Sterol Side-Chain Cleavage in Digoxin Biosynthesis" (NCOMMS-23-03663A). Below we address the comments and concerns of each reviewer.

Reviewer #1

- "Unfortunately, they only observed knockout mutants for use of campesterol and thus did not obtain information about how the CYP87A4 enzyme imposed differential substrate selectivity in comparison to the mammalian sterol cleavage enzyme."

Response: We thank the reviewer for raising this valuable point. In the original submission, we only tested the human P450_{scc} (CYP11A1) using cholesterol as the substrate but tested the plant CYP87A4 using five different substrates. To directly compare the substrate selectivity of these two P450_{scc}s, we additionally tested the human CYP11A1's selectivity for all five substrates. The five sterol-producing yeast strains were transformed with a plasmid expressing the human CYP11A1, adrenodoxin (ADX), and adrenodoxin reductase (ADR). Sterols are extracted and analyzed by GC/MS to detect the reaction product pregnenolone. Results showed the human CYP11A1 is specific to cholesterol whereas the plant CYP87A4 can take both campesterol and cholesterol as substrates. Figure 3 is updated to include the additional data as below.

Figure 3: Results from yeast *in vivo* assays. *D/CYP87A4* and human CYP11A1 were expressed in 5 different strains of yeast each expressing a main sterol; ergosterol, 7-dehydrocholesterol, desmosterol, campesterol, and cholesterol.

We also compared the active centers of the two P450_{scc}s to investigate how the different substrate selectivity arises. We identified amino acids within 4.5 Å of the docked cholesterol in the two protein models. The overall shape of the substrate binding pocket and the positioning of heme and cholesterol are similar in both P450_{scc}s (Figure 7A&B). Both

substrate recognition sites were comprised mainly of non-polar amino acids indicating hydrophobic interactions are the main driving force for substrate binding (Figure 7C). However, the amino acids that constitute the two active sites are distinct. Only two amino acids interacting with cholesterol are similar in these two P450_{scc}s. Specifically, L357 in CYP87A4 and V353 in CYP11A1 interact with C2 of cholesterol, and L212 in CYP87A4 and I84 in CYP11A1 interact with C7. Interestingly, mutating L357 in CYP87A4 abolished the side-chain cleaving activity (Figure 5D), supporting the critical role of this aliphatic amino acid in substrate recognition. We speculate that the distinct amino acid compositions of the human and plant P450_{scc} active sites underline the different substrate selectivity of the two enzymes.

Figure 7: Comparison of substrate recognition sites of A) *Digitalis lanata* CYP87A4 and B) Human CYP11A1. Cholesterol is docked to the active sites. Similar amino acids at near-identical positions of *DICYP87A4* and CYP11A1 are shown. C) List of active-site amino acids within 4.5Å to cholesterol in *DICYP87A4* and human CYP11A1. Green: similar amino acids in the two active sites; tan: hydrophobic amino acids in both active sites; gray: distinct amino acids.

We included this figure discussion section of the revised manuscript with the following text (Lines 297 to 301):

“Comparing the substrate recognition sites of DIP450_{scc} and the human P450_{scc} revealed that the amino acids are distinct, although both sites are comprised mainly of non-polar amino acids, indicating hydrophobic interactions are the main driving force for substrate binding (Figure 7).”

- “Overall, the work presented in this manuscript is well done and the identification of the essential sterol side chain cleavage enzyme is compelling. What isn’t so well done is the written text. Firstly, the manuscript itself should be written without necessity to consult the supplemental information about the experiments. For instance, what are the 6 treatment samples for leaf and root in the transcript profiling experiments? Duplicates of the biological replicates? Why weren’t leaf and root tissue differentiated by age to give a dynamic associated with digitoxin accumulation? A small item, but nonetheless illustrative of the writing style, the transcript profiling is about relative transcript abundance, not expression”.

Response: We are grateful to the reviewer for pointing out specific things that were not clearly written. The six treatment samples for leaves and roots were three biological replicates of

leaves or roots and two technical replicates for each sample. We've added the following sentence to the results section (Line 61 to 62):

"Total RNA from leaf and root tissues, including three biological replicates and two technical replicates from each tissue, were pooled to generate a reference transcriptome of *D. lanata*."

We've also modified the method section, "Plant material, RNA isolation, and sequencing" (Line 338 to 340), as follows:

"Leaf and root tissues from three different seedlings were used to prepare the Illumina sequence library. Each seedling represents one biological replicate, and total RNA from each replicate was split into two technical replicates."

We did not include leaf and root samples of different ages in the transcript profiling because no prior work analyzed the correlation between cardiac glycoside composition and the age of *D. lanata* seedlings. Thus, we were not sure what to expect if using samples from seedlings of different developmental stages. In addition, the *D. lanata* seeds germinate at different times (about 10 days to 21 days) even they were sowed at the same time in the same container. Accurately grouping plants according to their ages is difficult.

We have adopted the phrase "relative transcript abundance" instead of "expression" throughout the manuscript as the reviewer suggested.

- "In Figures 2 and 3, I'm very interested in how much of the available campesterol and cholesterol are converted to products. This is a measure of relative enzyme efficiencies and important to distinguish between a true enzymatic conversion versus a promiscuous activity."

Response: We agree that quantifying the specific catalytic activities of CYP87A4 will reveal if the enzyme prefers campesterol or cholesterol. Such experiment requires adding either campesterol or cholesterol to the CYP87A4 enzyme and its reductase, or cell lysate containing CYP87A4 and the reductase, followed by quantifying NADPH's turnover or pregnenolone formation. The yeast *in vivo* experiment in the manuscript cannot quantify the specific catalytic activities of CYP87A4 because the cells keep synthesizing the substrates NADPH and campesterol/cholesterol, and acetylating the product pregnenolone. Over the past two years, we have tried many times using yeast microsomes containing CYP87A4 and the reductase ATR2, but no pregnenolone was detected when adding campesterol or cholesterol. We have also expressed CYP87A4 in *E. coli* and incubated the cell lysate with purified redox partners but still didn't detect pregnenolone. We also tried to purify the N-terminal his-tagged CYP87A4 from *E. coli* lysate but the enzyme precipitated on the nickel column. In all these attempts the heterologously expressed CYP87A4 only absorbed at A_{420nm} instead of A_{450nm}, indicating the protein is inactivated once the cells were lysed. Below are our attempts to measure specific catalytic activities of CYP87A4 expressed in either yeast or *E. coli*:

Table 1: *In vitro* assay attempts using yeast microsomes.

Date	Attempt	Results	Changes
6/3/21	1	No pregnenolone seen	
9/20/21	2	No pregnenolone seen	Added DICYP87A4 to test. CYP87A4 was expressed along with Arabidopsis thaliana cytochrom P450 reductase, ATR2.

10/15/21	3	No pregnenolone seen	Repeat of attempt 2 with all 4 phytosterols (Campesterol, β -sitosterol, Stigmasterol, Cholesterol)
10/21/21	4	No pregnenolone seen	Straight repeat of attempt 3
10/26/21	5	No pregnenolone seen	Repeat of 3 with incubation times tested
11/23/21	6	No pregnenolone seen	Repeat of 3 with only cholesterol and incubation times tested
1/12/22	7	No pregnenolone seen	Plasmids changed to have inducible promoters for stronger protein expression. Human and DICYP87A4 on pGal1 and ATR2 under pCup1 promoter. Only cholesterol tested
1/14/22	8	No pregnenolone seen	Repeat of attempt 7 with other phytosterols (Campesterol, β -sitosterol, Stigmasterol, Cholesterol)
3/18/22	9	No pregnenolone seen	Repeat of 7 with a large scale (6L) preparation of yeast culture for larger protein expression in microsomes
4/12/22	10	No pregnenolone seen	Repeat of 7 with different ratio of human CYP11A1 to ADR and ADX
4/22/22	11	No pregnenolone seen	Switch back to plasmids with genes under constitutive promoters but with a large scale 6L preparation of yeast cultures
7/13/22	12	No pregnenolone seen	Buffer optimization with attempt 2 as base and using campesterol as substrate
7/21/22	13	No pregnenolone seen	Repeat of attempt 12 but with different incubation time

Table 2: *In vitro* assay attempts using *E. coli* cell lysate and purified protein.

Date	Attempt	Results	Changes
5/3/22	1	No pregnenolone seen	6L Preparation of Arctic E. coli expressing pTrcHisA-CYP87A4 plasmid cell lysate used for in vitro testing
5/10/22	2	No pregnenolone seen	Repeat of attempt 1 with incubation time tested
5/28/22	3	No pregnenolone seen	Repeat of attempt 1 with E. coli grown at different temperature
6/30/22	4	No P450 protein expression seen in protein preparation	3L protein purification

- “The text description of Fig. 2 also suggests that pregnenolone isn’t observed, but the actual figure suggests otherwise. A lot of these issues could be addressed by inclusion of quantitative assessments (how much substrate converted to product) and careful attention to the writing style”.

Response: We thank the reviewer for pointing this out. The minor peak in Fig. 2C set 1 is not pregnenolone since it has a slightly different retention time than the pregnenolone standard. A sentence is added in the results section (Line 142 to 143) to clarify:

“Note that the minor peak is not pregnenolone due to the different retention time compared to pregnenolone. “

Accordingly, a sentence is added in the figure legend to emphasize again (Line 157 to 158):

“Note the minor peak is not pregnenolone because of the different retention time compared to the pregnenolone standard.”

We tried many times to quantifying the catalytic activities of CYP87A1 but failed due to its inactivation *in vitro*. Please see our attempts in the Table 1 and 2 above.

We have carefully read through the revised manuscript and corrected any inconsistencies.

- *“They really do not address the issue that plants have evolved a sterol side chain cleavage enzyme unlike that in mammals. The upshot of this is these two kingdoms have evolved two very different classes of steroidal based hormones, one requiring a SCC cleavage enzyme, the other not. But that then provided a means for plants to evolve a SCC cleavage enzyme for yet another function, for cardenolide biosynthesis. And this is unique to very specific plant families.”*

Response: We are grateful to the reviewer for pointing out the functional distinctions between mammalian and plant P450_{scs}. We added the following sentence in the discussion section (Line 260 to 264):

“While mammalian P450_{scs} for steroid hormone biosynthesis are indispensable for normal development, plant P450_{scs} evolved to synthesize specialized metabolites which only present in very specific plant families.”

Reviewer #2

- *“From the point of gene discovery, the gene was not the first identified gene with the function and the digoxin biosynthetic pathway had been reported previously”.*

Response: We regret not clearly communicate the novelty of this work. This gene is the first identified gene with the campesterol and cholesterol side-chain-cleaving function. No other enzyme exhibiting identical function has been reported before. The only similar gene that cleaves the cholesterol side chain is the mammalian P450_{scs}, which has no obvious homology with the identified plant P450_{scs}, and which does not recognize phytosterols, such as campesterol. The catalytic mechanisms of the plant and the mammalian P450_{scs} may also differ, warrant future investigation. The reported digoxin biosynthetic pathway was based on the radiolabeling conducted in the 60's. The pathway will remain putative until each enzyme catalyzing every reaction is identified and characterized. The pathway requires at least nine enzymes, and only two was identified before this work. We identified the first and rate-limiting enzyme of this pathway for the very first time. This discovery also rewrote the pathway to add phytosterols as substrates for digoxin. At least six more enzymes remain unidentified in the digoxin biosynthetic pathway.

- *“Given that the industrial significance of the catalytic function of sterol side chain cleaving, it's equally important to get a higher activity of enzyme. We hope the author can compare the catalytic activity of DICYP87A4, DICYP87A4' homologs and other reported P450scs.”*

Response: We agree that quantifying the specific catalytic activities of DICYP87A4 and human CYP11A1 will reveal which enzyme has higher activity for cleaving cholesterol's side chain. DICYP87A4's homologues do not exhibit any sterol side-chain cleaving activity (Figure 5E).

Besides the mammalian P450_{scc}, no other P450_{scc} has been reported before this work. Quantifying specific activities requires adding cholesterol to the purified side-chain-cleaving enzyme and its reductase, or cell lysate containing the enzyme and the reductase, followed by quantifying NADPH's turnover or pregnenolone formation. The yeast *in vivo* experiment in the manuscript cannot quantify the specific catalytic activities of CYP87A4 because the cells keep synthesizing the substrates NADPH and campesterol/cholesterol, and acetylating the product pregnenolone. Over the past two years, we have tried many times using yeast microsomes containing CYP87A4 or CYP11A1 and the reductase ATR2, or ADX and ADR, but no pregnenolone was detected when adding campesterol or cholesterol. We have also expressed CYP87A4 in *E. coli* and incubated the cell lysate with purified redox partners, but still didn't detect pregnenolone. We also tried to purify the N-terminal his-tagged CYP87A4 from *E. coli* lysate but the enzyme precipitated on the nickel column. In all these attempts the heterologously expressed CYP87A4 only absorbed at A_{420nm} instead of A_{450nm}, indicating the protein is inactivated once the cells were lysed. Below are our attempts to measure specific catalytic activities of CYP87A4 expressed in either yeast or *E. coli*:

Table 1: *In vitro* assay attempts using yeast microsomes.

Date	Attempt	Results	Changes
6/3/21	1	No pregnenolone seen	
9/20/21	2	No pregnenolone seen	Added DICYP87A4 to test. CYP87A4 was expressed along with Arabidopsis thaliana cytochrom P450 reductase, ATR2.
10/15/21	3	No pregnenolone seen	Repeat of attempt 2 with all 4 phytosterols (Campesterol, β -sitosterol, Stigmasterol, Cholesterol)
10/21/21	4	No pregnenolone seen	Straight repeat of attempt 3
10/26/21	5	No pregnenolone seen	Repeat of 3 with incubation times tested
11/23/21	6	No pregnenolone seen	Repeat of 3 with only cholesterol and incubation times tested
1/12/22	7	No pregnenolone seen	Plasmids changed to have inducible promoters for stronger protein expression. Human and DICYP87A4 on pGal1 and ATR2 under pCup1 promoter. Only cholesterol tested
1/14/22	8	No pregnenolone seen	Repeat of attempt 7 with other phytosterols (Campesterol, β -sitosterol, Stigmasterol, Cholesterol)
3/18/22	9	No pregnenolone seen	Repeat of 7 with a large scale (6L) preparation of yeast culture for larger protein expression in microsomes
4/12/22	10	No pregnenolone seen	Repeat of 7 with different ratio of human CYP11A1 to ADR and ADX
4/22/22	11	No pregnenolone seen	Switch back to plasmids with genes under constitutive promoters but with a large scale 6L preparation of yeast cultures
7/13/22	12	No pregnenolone seen	Buffer optimization with attempt 2 as base and using campesterol as substrate
7/21/22	13	No pregnenolone seen	Repeat of attempt 12 but with different incubation time

Table 2: *In vitro* assay attempts using *E. coli* cell lysate and purified protein.

Date	Attempt	Results	Changes
5/3/22	1	No pregnenolone seen	6L Preparation of Arctic E. coli expressing pTrcHisA-CYP87A4 plasmid cell lysate used for in vitro testing
5/10/22	2	No pregnenolone seen	Repeat of attempt 1 with incubation time tested
5/28/22	3	No pregnenolone seen	Repeat of attempt 1 with E. coli grown at different temperature
6/30/22	4	No P450 protein expression seen in protein preparation	3L protein purification

- “The study revealed two amino acids which is critical for CYP87A’s catalyzation. We hope the author can discuss the similarities and differences of key residues in catalytic pocket between CYP87A and CYP11A1.”

Response: We thank the reviewer for suggesting this valuable analysis. We compared the key residues in the catalytic pockets of CYP87A and CYP11A1. We identified the amino acids within 4.5 Å of the docked cholesterol in the two protein models. The overall shape of the active centers and the positioning of heme and cholesterol are similar in both P450_{sccs} (Figure 7A&B). Both reaction sites were comprised mainly of non-polar amino acids indicating hydrophobic interactions are the main driving force for substrate binding. However, the amino acids that constitute the two active sites are distinct. Only two amino acids interacting with cholesterol are similar in these two P450_{sccs}. Specifically, L357 in CYP87A4 and V353 in CYP11A1 interact with C2 of cholesterol, and L212 in CYP87A4 and I84 in CYP11A1 interact with C7. Interestingly, mutating L357 in CYP87A4 abolished the side-chain cleaving activity (Figure 5D), supporting the critical role of this aliphatic amino acid in substrate recognition. We speculate that the uniqueness of amino acid compositions of the human and plant P450_{scc} active sites underlines the differential substrate selectivity of the two enzymes. This new figure is included into the revised manuscript.

Figure 7: Comparison of substrate recognition sites of A) *Digitalis lanata* CYP87A4 and B) Human CYP11A1. Cholesterol is docked to the active sites. Similar amino acids at near-identical positions of DICYP87A4 and CYP11A1 are shown. C) List of active-site amino acids within 4.5Å to cholesterol in

DICYP87A4 and human CYP11A1. Green: similar amino acids in the two active sites; tan: hydrophobic amino acids in both active sites; gray: distinct amino acids.

- “Line 57. CYP78A or CYP87A? ”

Response: Thanks for catching this typo. We corrected to “CYP87A” in Line 55.

- “Line 128. Therefore, hese two transcirpts., hese?”

Response: Thank you for catching this typo. We corrected it to “these” in Line 129.

Reviewer #3

“Finally, protein modeling, with substrate docking and comparison of these ‘extra’ CYP87A sub-family members from Digitalis relative to the generally single copies more widely found, identified two residues whose change in identity is important for the sterol side-chain cleavage activity. Unfortunately, it remains unclear what role the ancestral CYP87 sub-family members play in plant metabolism, so only loss of activity could be determined here. On the other hand, it might be of interest to determine if these two changes are sufficient to impart the ability to catalyze sterol side-chain cleavage.”

Response: We thank the reviewer for suggesting this valuable experiment. We mutated these two amino acids in the ancestral DICYP87A1 to the ones in the sterol side-chain-cleaving DICYP87A4, named DICYP87A1 V352A A354L. The double mutant was tested using the campesterol-producing yeast. However, no convincing pregnenolone peak was shown indicating these two amino acids are insufficient to impart the sterol side-chain cleavage and more amino acids are required to confer this function. Testing this double mutant in the other sterol-producing yeasts, including cholesterol, 7-dehydrocholesterol, desmosterol, and ergosterol, also showed no convincing pregnenolone peak. This data is included as Fig. S13 in the supplementary information and the results section in the manuscript (Line 237 to 239) as:

“However, these two amino acids are insufficient to impart the sterol side-chain-cleaving activity as the canonical DICYP87A1 mutated with these two amino acids was unable to cleave campesterol (Supplementary Figure S13).”

Figure S13: Extracted ion chromatogram for pregnenolone from the campesterol-producing yeast strain expressing various *Digitalis lanata* CYP87As and the ATR2 redox partner.

- “Ln 40: ‘works’ should be ‘work’
Response: Changed. Thank you.
- “Ln 64: performed in de novo should be performed de novo”
Response: Changed. Thank you.
- “5.66 Å 4.93” should be “5.66 Å and 4.93”
Response: Changed. Thank you.

Reviewer #4

- “Characterization of the first P450_{scc} in plant is an important step for deciphering complete pathway of cardiac glycosides. Experiments of this paper are generally well conducted. However, I think this is a routine story accomplished by common strategies starting from mining of RNA-seq data to functional identification. Only one p450 gene has been reported, which does not represent enough novelty suitable for acceptance.”

Response: We thank the reviewer for appreciating the importance and the rigor of this work. We regret not clearly communicate the novelty of the results. While the approach to identify the gene is standard, the P450 gene reported is the first identified plant sterol side-chain-cleaving enzyme. This enzyme is unlike most other P450s that monooxygenate their substrates. This P450 catalyzes a bond cleavage between unfunctionalized carbons presumably through three consecutive oxygenation reactions. Similar reactions were only reported in mammalian systems but never in plants. In addition, the catalytic mechanism of the enzyme reported may differ from its animal counterpart. Thus, this work lays the foundation for future mechanistic studies of plant P450_{scc}. Finally, the P450 reported is the only enzyme that cleaves the side chain of campesterol, a plant sterol, so far identified.

- “In Fig 1, please enlarge the chemical structures. The structure of isoprogestosterone is not correct.”

Response: We thank the reviewer for careful reading. The sizes of the structures of Figure 1 have been increased and the structure of isopregnenolone has been corrected in the revised manuscript.

- “I wonder if beta-sitosterol and stigmasterol can be served as the substrates of *DICYP87A4*, since these sterols are more generally present in plants.”

Response: We thank the reviewer for this insightful comment. We explored the possibility of β -sitosterol and stigmasterol as substrates for *DICYP87A4*. We initially tried to test this possibility *in vitro* using yeast microsomes or *E. coli* lysate expressing this enzyme and the redox partner, but none of the experiments worked despite many attempts (See Tables 1&2 above). We utilized docking simulations instead. We docked the *DICYP87A4* protein model with campesterol, cholesterol, β -sitosterol, and stigmasterol. We found the lowest energy orientations are the ones that docked the sterol side chains the closest to the heme center (Fig. 6). Docking of campesterol, cholesterol, and β -sitosterol put the C20 and C22 of the sterols within 4.6-5.6Å of the heme center (Fig 6A, B, and C). Note that the *DICYP87A4* cleaves the carbon bonds between C20 and C22. Since cholesterol and campesterol can be cleaved from this distance, it is likely that the β -sitosterol can also be cleaved too. However, docking with stigmasterol put the C20 and C22 over 7Å away from the heme center (Fig 6D). This leads us surmise that the stigmasterol is not be cleaved by this enzyme. Stigmasterol's C22:23 double prevents bond rotation and could result in the 24-ethyl group pointing towards the heme center (Fig 6D), preventing C20 and C22 from getting close.

Figure 6: Docking simulations of *DICYP87A4* with A) campesterol, B) cholesterol, C) β -sitosterol, D) and stigmasterol. Dashed lines show the distance of C20 and C22 to the heme center. Docking conformations shown are of the lowest energy in each simulation.

We included Figure 6 into the manuscript and the following sentence in the discussion (Line 273 to 281):

“In order to test if the other major plant sterols, β -sitosterol and stigmasterol, may also serve as substrates for $DIP450_{SCC}$, we performed docking simulations (Figure 6). While the C20 and C22 of campesterol, cholesterol, and β -sitosterol are all within 4.6-5.6Å to the heme center, docking with stigmasterol put the two carbons over 7 Å away from the heme. The 22:23 double bond of stigmasterol prevents bond rotation and results in the bulky 22-ethyl group pointing towards the heme center, preventing C20 and C22 from accessing the heme. Thus, we surmise that β -sitosterol could also be $DIP450_{SCC}$'s substrate, along with cholesterol and campesterol, but not stigmasterol.”

- “It is not clear why the cloning of CYP87A3 failed. Can has this ORF synthesized for test?”

Response: Following the reviewer’s suggestion, we synthesized CYP87A3 and expressed it in the campesterol-producing yeast strain. However, no clear and convincing pregnenolone peak was detected, unlike CYP87A4, as shown below.

Figure S13: Extracted ion chromatogram for pregnenolone from the campesterol-producing yeast strain expressing various *Digitalis lanata* CYP87As and the ATR2 redox partner.

This data is included as Figure S13 in the revised manuscript. We also included a sentence in the results section (Line 195 to 197):

“*DICYP87A3* is 97.4% identical at the protein level to the characterized *DICYP87A4* but cannot cleave the side chain of campesterol when expressed in yeast (Supplementary Figure S13).”

- “What are the parameters of tBLASTx search for CYP87A transcripts?”

Response: The parameters of the tBLASTx search for CYP87A were the default parameters in the 1,000 Plant Transcriptome database (<https://db.cngb.org/onekp/>). Genetic code: standard, strand: both. Database: 1000 plant project Database code: onekp, Project: oneKP, Format version v5. Transcripts were filtered, and only those within 400-600 amino acids long and had a start codon were retained. We modified the methods section (Lines 413 to 415):

“Transcriptome sequences used for the CYP87A tree were retrieved by BLASTing the *DICYP87A4* transcript against the 1,000 Plant Transcriptome (1KP) database using tBLASTx

^{36,50} under default parameters. Sequences were filtered, and only those within 400-600 amino acids long and had a start codon were retained.”

Respectfully,

Zhen Wang

April 20, 2023

Reviewer #1 (Remarks to the Author):

In the current revised manuscript, the authors have responded to previous reviewer criticisms well. I suggested that the discovery of this unusual sterol side-chain cleavage enzyme represented a more significant finding than the authors conveyed in the original manuscript. In response, the authors have relied upon a phylogenetic analysis to suggest how unique this particular CYP is and how narrow is its finding in one particular species of plants. All of this is good, but really what I was trying to suggest is that the evolution of this biosynthetic activity predates hormonal developmental programs in the Animalia Kingdom of life, and that its evolution in a specific plant species in Plantae Kingdom probably arose because of some selective pressure for this function beyond hormonal control of development. The current manuscript still does not quite capture this perspective, but this is a key finding of the work in this reviewer's opinion. Put a little differently, given that this is likely a key differentiation point in the Kingdom of Life branches between plants and animals, I would place the origin of these enzyme activities (and genes) at least 600 million years ago (a key branch point in the geological evolution of planet earth) for some survival functions differentiating plants and animals yet understood.

Several reviewers requested additional information about the knockout mutations of the CYP87A4. The knockout mutations are interesting but not without their limitation. A knockout, without other supporting evidence, is a negative result. They suggest that a residue is essential for enzyme function, but that does not mean catalysis. It could simply be that the mutant alters protein folding or protein stability. The way such mutants are qualified for measuring changes in actual enzyme activity in P450 is by complementing the activity measurements with spin states of the enzyme. This appears not to be possible in the current situation for technical reasons. The authors have not succeeded in converting other CYP87 enzymes by site-directed mutagenesis to enzymes with side-chain cleavage either. And this, in this reviewer's opinion, is strong evidence that the two amino acid positions may contribute to side-chain cleavage, but certainly are not sufficient for such.

Overall, the finding of this enigmatic enzyme activity is very significant and will have far reaching implications for a very diverse array of readers and investigators.

Reviewer #2 (Remarks to the Author):

The authors have resolved my concerns. I do not have more questions.

Response to Reviewer Comments

The authors thank the reviewers for thoughtful comments, which have been incorporated into the revised manuscript “A Cytochrome P450 CYP87A Imparts Sterol Side-Chain Cleavage in Digoxin Biosynthesis” (NCOMMS-23-03663A). Please find the point-by-point responses to the reviewer comments below:

Reviewer #1

- “In the current revised manuscript, the authors have responded to previous reviewer criticisms well. I suggested that the discovery of this unusual sterol side-chain cleavage enzyme represented a more significant finding than the authors conveyed in the original manuscript. In response, the authors have relied upon a phylogenetic analysis to suggest how unique this particular CYP is and how narrow is its finding in one particular species of plants. All of this is good, but really what I was trying to suggest is that the evolution of this biosynthetic activity predates hormonal developmental programs in the Animalia Kingdom of life, and that its evolution in a specific plant species in Plantae Kingdom probably arose because of some selective pressure for this function beyond hormonal control of development. The current manuscript still does not quite capture this perspective, but this is a key finding of the work in this reviewer’s opinion. Put a little differently, given that this is likely a key differentiation point in the Kingdom of Life branches between plants and animals, I would place the origin of these enzyme activities (and genes) at least 600 million years ago (a key branch point in the geological evolution of planet earth) for some survival functions differentiating plants and animals yet understood.”

Response: We deeply appreciate the reviewer’s valuable perspective, which would place this original discovery in a significantly larger context in evolution than the current manuscript suggests. The reviewer suggests that the mammalian and *Digitalis lanata* P450_{sccS} originated from a common ancestor more than 600 million years ago (mya) before plants and animals split in the Tree of Life. Indeed, two possibilities exist: 1) The animal and plant P450_{sccS} originate from a common ancestor before the split of plant and animal kingdoms, an exciting divergent evolution hypothesis; 2) The sterol cleavage activity (P450_{scc} gene) independently originated more than once at different time points in animals and plants, a convergent evolution hypothesis.

For divergent evolution to happen, all plant species, except a few cardenolide-producing ones, must have lost the ancestral sterol cleavage activity during evolution. Since there are ~320,000 plant species and only about ~12 can produce cardenolides, the loss-of-function events must have happened at least thousands of times independently in evolution, a statistically extremely unlikely event. Meanwhile, we would expect the CYP87A genes in most non-cardenolide-producing plants to accumulate loss-of-function mutations such as internal stop codons, frameshifts, or transposon insertions, thus producing no transcripts. However, CYP87A transcripts are abundant in non-cardenolide-producing plants. In fact, the phylogenetic tree in Figure 4 was built using CYP87A transcripts from different plants.

In contrast, the convergent evolution hypothesis would be the most parsimonious or the most straightforward explanation and agrees with the data collected. Under this hypothesis, the ~12 plant species gained the P450_{sc} activity independently under certain selective pressure (defense). These ~12 independent events happened after the split of plants and animals. These gain-of-function events only need to happen roughly a dozen times in ~600 mya, which is statistically possible. For foxglove in particular, we think the P450_{sc} likely evolved after the foxglove split from a closely related species, *Antirrhinum majus*, which produces no cardenolides, about ~48 mya. Mechanistically, these gain-of-functions require gene duplications to free one of the gene copies to evolve a new function (neofunctionalization). This is the case for *D. lanata*'s CYP87A. While most other plants, including the closely-related snapdragon, have only one copy of CYP87A, *D. lanata* has four. Only the CYP87A4 has the P450_{sc} activity (Figure 5, Supplementary Figure 13). Thus, the authors think that the most plausible explanation for the evolution of P50_{sc} would be the gain of function in several cardenolide-producing plants rather than a loss of function in most plants.

- “Several reviewers requested additional information about the knockout mutations of the CYP87A4. The knockout mutations are interesting but not without their limitation. A knockout, without other supporting evidence, is a negative result. They suggest that a residue is essential for enzyme function, but that does not mean catalysis. It could simply be that the mutant alters protein folding or protein stability. The way such mutants are qualified for measuring changes in actual enzyme activity in P450 is by complementing the activity measurements with spin states of the enzyme. This appears not to be possible in the current situation for technical reasons. The authors have not succeeded in converting other CYP87 enzymes by site-directed mutagenesis to enzymes with side-chain cleavage either. And this, in this reviewer’s opinion, is strong evidence that the two amino acid positions may contribute to side-chain cleavage, but certainly are not sufficient for such.”

Response: The reviewer correctly pointed out the limitations of the knockout mutations of CYP87A4, with which we fully agree. We added the following sentences to the manuscript:

In Discussion, lines 386-397: “*However, since the activity assay used cell lysate instead of purified proteins, which are difficult to isolate despite repetitive attempts, we do not rule out the possibility that these two mutations may also affect protein folding or stability.*”

In Results, lines 274-276: “*However, these two amino acids are insufficient to impart the sterol side-chain-cleaving activity as the canonical DICYP87A1 mutated with these two amino acids was unable to cleave campesterol.*”

- “Overall, the finding of this enigmatic enzyme activity is very significant and will have far reaching implications for a very diverse array of readers and investigators.”

Response: We are grateful to the reviewer’s appreciation of this work.

Reviewer #2

- “The authors have resolved my concerns. I do not have more questions.”
Response: We appreciate it.